# Engineering Therapeutic Strategies in Cancer Immunotherapy via Exogenous Delivery of Toll-like Receptor Agonists

**DOI:** 10.3390/pharmaceutics13091374

**Published:** 2021-08-31

**Authors:** Sehwan Jeong, Yunyoung Choi, Kyobum Kim

**Affiliations:** Department of Chemical & Biochemical Engineering, Dongguk University, 30, Pildong-ro 1-gil, Jung-gu, Seoul 22012, Korea; storyjsh@gmail.com (S.J.); truejy0e@gmail.com (Y.C.)

**Keywords:** TLR agonist, adjuvant, cancer immunotherapy, drug delivery system, nanomedicine

## Abstract

As a currently spotlighted method for cancer treatment, cancer immunotherapy has made a lot of progress in recent years. Among tremendous cancer immunotherapy boosters available nowadays, Toll-like receptor (TLR) agonists were specifically selected, because of their effective activation of innate and adaptive immune cells, such as dendritic cells (DCs), T cells, and macrophages. TLR agonists can activate signaling pathways of DCs to express CD80 and CD86 molecules, and secrete various cytokines and chemokines. The maturation of DCs stimulates naïve T cells to differentiate into functional cells, and induces B cell activation. Although TLR agonists have anti-tumor ability by activating the immune system of the host, their drawbacks, which include poor efficiency and remarkably short retention time in the body, must be overcome. In this review, we classify and summarize the recently reported delivery strategies using (1) exogenous TLR agonists to maintain the biological and physiological signaling activities of cargo agonists, (2) usage of multiple TLR agonists for synergistic immune responses, and (3) co-delivery using the combination with other immunomodulators or stimulants. In contrast to naked TLR agonists, these exogenous TLR delivery strategies successfully facilitated immune responses and subsequently mediated anti-tumor efficacy.

## 1. Introduction

Cancer immunotherapy has been intensively and extensively investigated to develop more effective cancer treatments. Unlike conventional therapeutic approaches, such as chemotherapy and radiotherapy, cancer immunotherapy potentially activates the patient’s immune system to eradicate cancer cells [1,2,3]. Cancer immunotherapeutic agents should be designed to provoke a robust primary and secondary antitumor immune response by repairing or enhancing natural mechanisms that during disease progression are evaded or damaged, thus inhibiting tumor growth and subsequent metastasis [4,5,6].

Among a series of antigen-presenting cells (APCs), dendritic cells (DCs) are the most important immune cells for anti-cancer effects [7,8], due to their characteristic functionality of gathering and processing antigens to present immunogenic epitopes. As a consequence of antigen presentation by DCs, cytotoxic T lymphocyte (CTL) activation against tumor cells and virus-infected cells and the expansion of activated natural killer (NK) cells to kill malignant tumor cells could be achieved in the immune system [9,10,11]. However, for effective antigen presentation capability, the maturation process of DCs is crucial, and the maturation process could modulate the phenotypic features of DCs. For example, mature DC-derived exosomes proved to be enriched in MHC class II, B7.2, and ICAM-1, while depleted in MFG-E8, providing that exosomes from mature DCs were (50 to 100)-fold more potent than exosomes from immature DCs, both in vitro and in vivo [12,13]. Moreover, the degree of DC maturation could also alter the efficiency of electroporation-based nonviral antigen (i.e., tumor messenger RNA) transfection into DCs [14], and enhance immunogenic capability, as compared with the tolerogenicity of partial or semi-maturation of DCs [15].

Toll-like receptor (TLR) is a pattern recognition receptor (PRR) protein that plays an important role in the immunogenic signal transduction of APCs, such as DCs and macrophages, via specific recognition with TLR agonists. Among the 10 different types of human TLRs reported, TLRs 1, 2, 4, 5, 6, and 10 are expressed on the cell surface, and recognize microbial membrane components, such as peptidoglycan or lipopeptides of bacteria, while TLRs 3, 7, 8, and 9 are located on endosomal membrane and recognize the DNA and RNA of virus or bacteria [16,17,18]. TLR agonist could be both a type of pathogen-associated molecular pattern (PAMP) signal that is expressed in many microbes (i.e., agents to perpetuate the inflammatory response to infection), as well as endogenous damage-associated molecular pattern (DAMP) proteins released by dying cells (i.e., host biomolecules that can initiate a noninflammatory response to infection) [19]. When TLRs recognize their corresponding TLR agonists, they recruit adaptor proteins, such as MyD88, TRIF, and TIRAP/MAL, to activate signaling pathway [20]. These adaptor proteins sequentially activate transcription factors to secrete cytokines/chemokines, type I interferons, and induce DC maturation. This DC maturation allows DCs to express CD80 and CD86 molecules. Activated DCs affect T lymphocyte to Th1 cells and produce IFN-γ, and promote B cell activation. They can also induce the differentiation of T lymphocyte to cytotoxic T lymphocyte (CTL) effectors [21,22,23,24].

Although TLR agonists can activate immune cells via recognition by TLR, some small molecules often fail to demonstrate satisfactory in vivo immune response. For example, unavailable pharmacokinetic profiles of some TLR agonists in the dynamic body environment [25], or easy dissipation and low accumulation in a lymph node due to the small size of TLR agonists, might reduce the chance of contact with immune cells, cause insufficient threshold for immune cell activation, and thereby hinder the effectiveness of the downstream signal transduction of TLR–TLR agonist recognition [26]. Rapid diffusion and dissipation from a localized site can also lead to the low therapeutic dose level of TLR agonists, with unwanted cytokine surges, and severe adverse immune-toxicity effects occurring by the agonist [27]. Another problem of exogenous TLR agonist delivery is the shortness of half-life in the body. In particular, RNA/DNA agonists for endosomal TLR stimulation are easily decomposed by nucleases in the body [28]. Therefore, to tackle such technical limitations of exogenous TLR agonist delivery, functional delivery platforms should be designed, fabricated, and used. To overcome these issues, nanoparticles, self-assembly substances with TLR agonists, and hydrogels of suitable size range could complement the lymph node accumulation of TLR agonist and efficiently activate immune cells [29,30]. A suitable size of nanoparticle can allow TLR agonists to drain to the lymphatic system, preferentially internalized by DCs, and can stimulate CD8 T cells well. Additionally, using a delivery platform can protect the bioactivity of TLR agonists from the harsh environment of the body, while continuously stimulating the immune cell through a sustainable release.

To this end, the present review summarizes the use of the biomaterial-based delivery platforms for effective exogenous TLR agonist delivery and associated synergistic biomedical advantages, and particularly emphasizes (1) current strategies in exogenous TLR agonist delivery to stimulate immune cells, (2) positive effects of delivering multiple TLR agonists, and (3) synergistic efficacies of various stimulatory co-factors with TLR agonists. On top of the therapeutic functions of TLR agonists in cancer immunotherapy, an optimal cargo combination of TLR agonists with other stimulatory factors and associated administration strategies could be an important parameter for the development of next-stage off-label medications.

## 2. Promoting Immune Responses through TLR Agonist Delivery

### 2.1. Strategies for Immune Cell Stimulation via Exogenous TLR Agonist Delivery

To enhance the efficacy of TLR agonists on immune responses, a variety of delivery platforms have been developed to maintain the bioactivity of TLR agonists, and facilitate their sustained release. (A summary of the representative examples of TLR agonist delivery platforms is shown in Figure 1.) Table 1 demonstrates biomaterial-mediated delivery platforms for exogenous single TLR agonist delivery, associated cellular mechanism in immune systems, and technical improvements by using these delivery platforms, according to the type of TLR agonists.

For example, Pawar et al. designed the RNA/DNA hydrogel (RDgel)-based delivery platform to increase TRL7/8 stimulation [31]. In their study, RDgel was fabricated by mixing hexapodRD6-1 and -2 due to the presence of complementary ends to one another. HexapodRD6 consists of oligoribonucleotide (ORN) of Guanosine- and uridine-rich single-stranded RNA (GU-rich RNA) and oligodeoxyribonucleotide (ODN) that is used to load ORNs into hexapodRD6. Through the gradual degradation of RD gels, GU-rich RNA could be released in a sustained manner from hydrogel, and contribute to high immunostimulatory activity. Based on the release of nucleotide-based TLR agonists, RDgel exhibited an effective anti-tumor efficacy on colon 26 murine colon carcinoma cells by activating RAW264.7 murine macrophage-like cells and DC2.4 cells. Additionally, RDgel induced the apoptosis route of cancer cells.

Several studies have used R848 (Resiquimod) TLR7/8 agonists (i.e., FDA-approved safe imidazoquinoline compound). As stated above, a polymeric particulate system could be one of the functional exogenous TLR agonist delivery platforms. Chen et al. suggested nanoparticles (NPs) fabricated using hydrophobic polyaniline (PANI)-conjugated hydrophilic glycol-chitosan (GCS) backbone for R848 delivery to the tumor environment [32]. A high level of hydrophobic interactions of PANI side chains induced NP formation, and R848 was subsequently loaded by the co-solvent evolution. Positively charged R848-incorporated NPs were successfully internalized into DCs, and the cargo R848 was gradually released as the decomposition of GCS backbone was progressed in a physiological condition [33]. Additionally, PANI side chains with high photothermal conversion efficiency could induce hyperthermia at a relatively low temperature in the local site. This increased the expression of DAMP molecules, indicating synergistic anticancer effects with a loaded TLR agonist R848. In in vitro test, R848-loaded NPs could efficiently promote DC maturation along with higher expression levels of CD80 and CD86, and the secretion of IL-6 and TNF-α increased over time. R848-loaded NPs significantly inhibited tumor growth, and enhanced the proliferation of anti-tumor effector T cells in BALB/c mice. Additionally, in the tumor microenvironment, the secretion of proinflammatory cytokine IL-6 increased, while the secretion of immunosuppressive cytokine IL-10 was downregulated. The growth of tumors was eventually inhibited by R848-incorporated polymeric NPs while facilitating long-lasting memory of immune responses.

In another study, semiconducting polymer nanoadjuvant (SPN_II_R) was developed using a semiconducting polymer nanoparticle core as an NIR-II photothermal converter doped with a R848 (TLR7/8 agonist) with a thermally responsive lipid (i.e., DSPE-PEG and DPPC) shell coating (Figure 1A) [34]. Once the lipid shell of the particle melted at about 42 °C, SPN_II_R-induced temperature reached 51 °C after laser irradiation for a suitable photothermal effect. In in vivo treatments using BALB/c mouse model, SPN_II_R-injected group with laser irradiation eradicated tumor tissues, while also effectively inhibiting the growth of distant tumors and pulmonary metastasis nodules. In addition, the population of matured DCs (CD80+ CD86+) increased, and subsequently the populations of activated T cells (CD3+CD4+ and CD3+CD8+) in both distant tumor and blood increased. Furthermore, the serum levels of immune-relevant cytokines, including interleukin-6 (IL-6), tumor necrosis factor-α (TNF-α), and interferon-γ (IFN-γ), were also up-regulated.

Similarly, imidazoquinoline-based small molecule agonist (termed 522) could be also incorporated into poly(lactic-co-glycolic acid) (PLGA) NPs by oil-in-water emulsion solvent evaporation [46]. The PLGA slowed the decomposition of agonists, and was effectively internalized by bone marrow (BM) DCs. After injecting PLGA NPs into C57BL/6 mice, accumulated 522-loaded PLGA NPs in lymph node could facilitate DC activation and expansion. Vaccination using 522-loaded PLGA NPs to C57BL/6 mice also increased the proliferation of antigen-specific CD8+ T cells, enhanced cytotoxic T lymphocytes (CTL) responses, and effectively hindered tumor growth in MB49 bladder cancer. Furthermore, a stimuli-responsive NP platform has also been investigated to facilitate the antigen presentation of DCs and downstream immune cell activation (Figure 1C) [47]. For example, sodium bicarbonate incorporated in PLGA NPs could produce CO_2_ gas in response to acidic pH condition in tumor microenvironments. Subsequently, accelerated disruption of NP structure enhanced the release of loaded 522 agonist, and improved endo-lysosome (pH 4–6)-specific 522 agonist release. Therefore, enhanced antigen uptake into DCs stimulated antigen presentation of the activated DCs, increased the proliferation of antigen-specific CD8+ T cells, and induced strong CTL response and NK cell activation.

Activated DC also increased the secretion of pro-inflammatory cytokines (IL-2, IL-15) and regulators for NK cell activation, such as IL-12p70 and IL-18. They increase the activation marker (CD25, CD69) and IFN-γ secretion of NK cells, and activate NK cells. To evaluate the effectiveness of NP as a vaccine adjuvant, NP was treated with cetuximab for A549 cell. NP treatment increased degranulated NK cells and IFN-γ secretion, and revealed more enhanced antibody-dependent cellular cytotoxicity (ADCC) for A549 cell [36].

Another TLR agonist used in delivery is the TLR 9 agonist, Cpg ODN. CpG is found in the DNA of some bacteria and viruses, and can bind to TLRs to trigger an immune response [49], but it is easily disintegrated by nucleases, thus increasing the half-life of CpG using synthetic ODNs is required [28]. Because CpG ODN is still sensitive to nuclease degradation, Liu et al. used dual-responsive polyamidoamine (PAMAM) Cluster for nanoadjuvant to deliver CpG ODN (Figure 1B) [35]. This nanocarrier enabled rapid and sustained release of CpG ODN in the extracellular tumor acidity and reductive environment of the endolysosome (Figure 2). Nanoadjuvant-activated BDMC increased the expression levels of IL-6 and IL-12. The combination of doxorubicin results in an induced immunogenic cell death in the tumor, and an increased CD8+ T cell number. Anti-PD-1 checkpoint blockade with nanoadjuvant reduces tumor growth, and shows a signature survival benefit.

In other cases, a study used hydrogel as a delivery platform. Perry et al. developed particle replication in non-wetting template (PRINT) hydrogel particles (Figure 1E) [38,48]. The amine handles on the NPs were used to conjugate thiol-modified CpG ODN to the particles through a heterobifunctional crosslinker, resulting in a thioether linkage. Negatively charged, monodispersed PRINT hydrogel particles promoted substantial tumor regression, and limited systemic toxicities. Mice were injected with lung cancer, and after treatment were cured throughout, to be completely resistant to tumor rechallenge. In addition, PRINT hydrogel extended the retention of CpG ODN, and accordingly prolonged the elevation of antitumor cytokines in the lungs, but with no elevation of the levels of pro-inflammatory cytokines in the serum.

For successful in vivo vaccination, the localization and accumulation of TLR agonist in lymph nodes should be initially achieved. In particular, in the case of oligonucleotide CpG TLR9 agonist delivery, the physiological protection of the cargo molecule and enhanced level of lymph node targeting could be obtained by conjugating amine-modified CpG ODN (CpG 1826) and oxidized dextran by reductive amination [29]. Without any compromising effect on the adjuvant activities of CpG agonist, enhanced CD8+ T cell responses and tumor suppression were observed in EG7 mice, with the aid of ovalbumin antigen. Moreover, the presence of long-term memory T cells efficiently inhibited tumor recurrence.

### 2.2. The Role of TLR Agonists in Immune Check Point Pathways

As depicted in Figure 3, the major role of TLRs in tumor immunity and immune check point pathways is activating DCs and subsequent upregulation of antigen presentation [50]. In addition to the recognition of PAMPs or DAMPs, DCs can present tumor-specific antigen (TSA) and tumor-associated antigen (TAA) from various tumor sources, and directly induce the differentiation of CD8+ T cells into cytotoxic T lymphocytes (CTLs). Through the interplay with MHC class I-bound antigen and T cell receptor (TCR) of CTLs in tumor microenvironments, CTLs can recognize and kill cancer cells. TLR agonists with anti-tumor efficacy could stimulate DCs, promote Th1-type immune responses, and activate tumor specific T cells [51] However, this CTL’s efficacy might be inhibited by PD-1/PD-L1 and CTLA-4 blockade [52,53].

Therefore, recent investigations have also focused on the function of TLR agonists as adjuvants to improve the efficacy of immune checkpoint inhibitors [50,54]. In general, a suggested mechanism for adjuvant effects of TLR agonists is activating DCs [55], increasing efficacy of anti-CTLA-4 antibody [56], activating CD8+ T cells [57], M1 macrophages [58] and NK-cells [59], decreasing M2 macrophages and Tregs inside the tumor microenvironments [60,61], downregulating PD-1 expression on CD8+ T cells [62,63], and escaping anti-PD-L1 resistance [64]. In addition to inducing antitumor responses in the innate immunity, generating adjuvant effects along with priming the adaptive immune response elicited by checkpoint blockade during tumor-cell killing phases could be obtained. Moreover, recent studies of combination therapies with TLR agonists and antibodies for immune checkpoint blockade have proven its potential usages and clinical applications [65,66,67,68].

### 2.3. TLR Agonists as Adjuvants for Cancer Vaccines

Since the early-stage therapeutic cancer vaccine, Provenge (sipuleucel-T), was approved by US Food and Drug Administration (FDA) for the treatment of androgen-independent advanced metastatic prostate cancer [69], there has been a series of investigations to develop effective adjuvants in order to overcome limitations of conventional cancer vaccines such as a poor immunogenicity and inappropriate immune responses [70]. Here, the adjuvant is a compound that directly engages the immune system and increases the potency of the antigen-specific immune responses via co-administration with cancer vaccine. In particular, TLR agonists could be also involved in a facilitated immunological presentation of antigens by DCs to induce subsequent T cell responses [71]. So far, only a limited number of TLR agonists have been licensed by the FDA for usage in human cancers, including Bacillus Calmette-Guerin (BCG, TLR2/4 agonist for in situ bladder cancer vaccine), monophosphoryl lipid A (MPLA, TLR2/4 agonist as an immunostimulatory adjuvant of Cervarix), and imiquimod (TLR7 agonist for malignancies) (Table 2) [71,72,73,74,75,76,77,78]. Although the therapeutic potentials of TLR agonists as adjuvants have partially been proven, advanced combination therapies and preclinical tests using exogenous delivery of multiple TLR agonists are also currently under investigation for consistent rates of remission, limited side effects, and higher immunostimulatory functions [72].

## 3. Combined Delivery of Multiple TLR Agonists for Enhanced Cancer Immunotherapeutic Efficacy

In general, a signaling event initiated by agonist recognition by TLR activates DCs, upregulates a secretion of pro-inflammatory cytokines, and sequentially promotes innate and adaptive immune responses. As compared with a single delivery of TLR agonist, exogenous delivery of multiple TLR agonists to DCs could simultaneously trigger various signaling pathways, and affect paracrine immunologic interplays, resulting in synergistic immune responses, such as facilitated T cell activation, and enhanced cytokine releases [79,80,81]. (Table 3 defines the targeted cancer and delivery platform according to TLR agonist.)

One feasible example of the combined delivery of various TLR agonists is the usage of lipo-immunogen for TLR2 targeting with a conventional CpG ODN TLR9 agonist [39]. Shen et al. reported the synergistic efficacy of immune activation as well as tumor inhibition via the combined delivery of lipodated human papillomavirus (HPV) E7 inactive mutant (rlipoE7m) as TLR2 agonist and phosphodiester CpG ODN with the aid of 1,2-Dioleoyloxy-3-trimethylammonium propane (DOTAP)-based cationic liposomes (Figure 1F). In terms of a delivery platform to encapsulate dual agonists, DOTAP could construct a stable complex with rlipoE7m TLR 2 agonist, and negatively charged DNA cargo as well. This liposomal complex could effectively activate both BMDCs and plasmacytoid DCs, and increase the in vitro production of IL-12p70. Moreover, after intravenous immunization in C57BL/6 mice, the following facilitated in vivo immune responses were observed, as compared with any single delivery of a cargo agonist: (1) downregulation of DC-secreted immunosuppressive IL-10, (2) significant reduction in tumor-infiltrating regulatory T cells, (3) enhanced activation of CTLs, and (4) effective inhibition of tumor growth.

To overcome clinical limitations (i.e., low efficacy and severe toxicity) via TLR agonist delivery, which could be caused by a rapid systemic diffusion of agonists to nontarget tissues, multicomponent TLR agonist assembly has recently been suggested [82]. Manna et al. reported supramolecular multivalent heterodimer agonists, which covalently linked cell surface targeting TLR 2/6a peptide and an endosomal active TLR 7/8a small molecule (Figure 4). With the aid of nonimmunogenic amphiphile sugar poly (orthoester) scaffold (OL-DSPOE), the resulting nano-assembly micelle structures effectively stimulate transcription factor NF-κB activity and improve the secretion of IL-12p70 and IFN-β of in vitro RAW Blue macrophages. After injection three times of this multicomponent nano-assembly (i.e., 9, 15, and 21 days after tumor inoculation) into a mouse model, sufficient antitumor efficacy (i.e., increased numbers of CD8+ T cells and NK cells, and reduced tumor growth) and mitigated off-target toxicity (i.e., normal blood cell viability and maintained spleen size) were observed. In particular, the system cytokine secretion post injection of dual agonists (i.e., TNF-α and IL-6 in blood serum) was not increased, as compared with 2/6a + 7a unlinked agonists and 2/6_7a-linked heterodimer, indicating more suitable localization of nano-assembly for dual agonist delivery.

Furthermore, a triple combination of TLR agonists could be efficiently delivered by using biomaterial-mediated particle platforms [37,83]. As mentioned above, soluble or hydrophilic small molecule adjuvants could be easily dissipated from the administration site, limit therapeutically effective dose in physiological body conditions, and reduce vaccine responses. To handle possible adverse immune-toxicity effects by this undesired diffusional phenomenon, PLGA-based polymeric pathogen-like particle (PLP) formulation has also been developed for incorporating Pam3CSK4 (TLR 1/2 agonist), monophosphoryl lipid A (MPLA; TLR 4 agonist), R837 (TLR 7 agonist), and CpG (TLR 9 agonist) (Figure 1D). Such combinatorial vaccination efficiently promoted in vitro antigen cross-presentation of BMDCs, by showing increased IFN-γ secretion in the presence of soluble ovalbumin during the co-culturation with OTI CD8 T cells. In addition, antigen-specific adaptive in vivo immune responses and enhanced humoral responses were also observed when mice were subcutaneously immunized with this triple combination of TLR agonists. Moreover, complement component 3 (C3) protein-associated liposomal formulation has been fabricated for targeted APC activation in the absence of specific tumor antigens (Figure 1G) [40,84], due to the priming efficacy of C3 through the complement receptors: (1) active encapsulation of foreign particles, and (2) increased uptake into myeloid cells [40]. A combinatorial delivery of triple TLR agonists of MPLA (TLR 4 agonist), R848 (TLR 7/8 agonist), and CpG 1826 (TLR 9 agonist) using complement C3-liposomes also enhanced monocyte activation, up-regulated the inflammatory cytokines gene expression (e.g., IRF7, IP-10, IL-1b, IL-6, IL-12, and TNF-α) in myeloid cells, and reduced in vivo murine tumor growth.

MPLA and CpG activate DCs with synergistic effects through two distinct pathways. synthetic high-density lipoprotein (sHDL) was used to boost the immune response of these two agonists (Figure 1K) [44]. As a result of delivering MPLA and CpG together through sHDL, it increases the expression levels of CD80, CD86 and IL-12p70 when compared with individually delivered groups. These results demonstrate that the delivery of dual TLR agonists through sHDL could significantly activate DCs. When OVA (i.e., antigen) was embedded with dual TLR agonists loaded sHDL, a high anti-OVA IgG titer was induced by a strong antibody response. Consequently, this TLR agonist and antigen multiple-delivery platform enhanced antigen-specific CD8+ T cell responses as well as antigen-specific CTL responses. Therefore, sHDL would be used as an effective immunologic adjuvant delivery platform to enhance anti-tumor efficacy.

## 4. Combinative Delivery of TLR Agonists with Various Biochemical Co-Factors Inducing Immune System

Regarding TLR agonists, other types of diverse substances boosting antigen activities, such as tumor cell lysates, recombinant protein, and fully synthetic antigenic, have also been used for TLR agonist-mediated cancer immunotherapy. In addition to facilitated immune activation via the exogenous delivery of dual/triple TLR agonists, potential benefits of the co-delivery of other and adjuvant substances have been investigated. Such co-adsorption or co-encapsulation at molecular scale and nano-scale delivery structures using polymeric scaffolds or inorganic templates could obtain higher possibility for the internalization of immune signals to generate more potent responses [85,86], and improved immunotherapies [87,88,89]. (Table 4 summarizes technical advantages of combinative delivery of TLR agonists with various co-factors.) 

### 4.1. Co-Delivery Using a Single Immune Substance

Among all kinds of TLR agonists, TLR 7 or 9 agonists seem to be the most selected inducers.

In the case of TLR7-mediated co-delivery (1-benzyl-2-butyl-1H-imidazo [4,5-c]quinolin-4-amine (BBIQ; TLR 7 agonist) and 1-methyl-D-tryptophan (D-1MT; also known as indoximod), higher stability and long shelf life non-ionic surfactant vesicles (niosomes) could be a compatible platform, offering higher stability and long shelf life (Figure 1H) [41]. Niosomes form closed bilayer structures with non-ionic surfactant membrane having aqueous core. BBIQ is expected to induce the production of IFN- α, and D-1MT has the prospect of targeting IDO pathways. Additionally, the transport mechanism of drug release for niosomes followed Fickian diffusion in nature.

Additional functionality from the intrinsic property of delivery platform materials could also elicit beneficial efficacy. Co-delivery of imiquimod TLR 7 agonist and anti-CD47 antibody using nanoscale metal–organic frameworks (nMOFs) could facilitate a radiotherapy–radiodynamic therapy (RT–RDT), based on the photodynamic and photosensitizing functions of nMOFs (Figure 1L) [45]. This formulation effectively induced macrophage repolarization by a combination of TLR 7 activation (immune modulation), CD47 blocking (inhibition of tumor cell-mediated phagocytosis), and immunogenic cell death by RT–RDT. With anti-PD–L1 immune checkpoint blockade, complete suppression of both primary and distant tumors on a bilateral colorectal tumor model was eventually confirmed as well. In addition, pH-responsible block copolymer RGD-PEG-b-PGA-g-(TETA-DTC-PHis) (RPTDH) is a R848 (TLR 7/8 agonist) delivery platform that releases its cargo molecule effectively in cancer environments (Figure 5) [90]. R848 activates immune cells, and subsequently gives the ability to suppress tumor growth and tumor metastasis. Meanwhile, RPTDH could also act as an inhibitor of cancer progression by chelating copper, which is a crucial factor in tumor angiogenesis PLP. To be specific, triethylene-tetramine-bis(dithiocarbamate) (TETA-DTC) acts as a copper chelator via amide bond, and poly-L-histidine (PHis) serves as a pH-sensitive polymer connected at the end amino group of TETA-DTC.

Furthermore, peptide-based antigen could also be incorporated for TLR7 agonist delivery. Tsai et al. reported a cancer vaccination strategy using polyplex-like nanoparticles condensed with human melanoma peptide antigen (Trp2, SVYDFFVWL) and anionic CPG (TLR 9 agonist) (Figure 1I) [42]. The delivery of tumor antigens and TLR ligand (i.e., Trp2R9:CpG) using polyplex could facilitate high-density co-display of antigen and adjuvant for strong, specific immune responses, with cargo protection. Treatment with Trp2R9:CpG polyplexes significantly increased the expression of surface activation markers CD40, CD86, and CD80 in DC cells. In addition, Trp2R9:CpG polyplexes displayed enhanced levels of IFN-γ, which is similar to dose-matched soluble Trp2R9 control in Trp2-specific CD8+T cells. Interestingly, modification of Trp2 peptide with arginine (1) could regulate absolute yield and loading of antigens on electrostatic condensation, while (2) dendritic cellular uptake of this arginine-modified Trp/CpG polyplex was dependent on the number of arginine residues. Hence, activated DCs presenting Trp2 peptide antigen with up-regulated TLR9 signaling successfully stimulated naïve T cell into Trp2-specific CTLs, and inhibited tumor growth with increased survival time in B16–F10 cells inoculated in murine in vivo model.

Similarly, Shi et al. developed a nanocomplex vaccine for the co-delivery of cell-penetrating peptide conjugated epitope and TLR9 agonist CpG (Figure 1J) [43]. This nanocomplex could be fabricated by the co-assembly of cationic epitope and CpG, and effectively activated immature host BMDCs. In addition to the increased stability of CpG in lymph nodes via nanocomplex formation, augmented CTL responses and memory T cell response were observed after subcutaneous immunization. Additionally, increased protective efficacy from B16 tumor challenge and subsequent anti-tumor effect were also obtained (Figure 6). More interestingly, this co-delivery strategy via peptide-mediated immunogenicity synergistically enhanced therapeutic efficacy via a PD-1 blockade approach.

Small-interfering ribonucleic acids (siRNA) are other co-delivery substances for enhanced TLR agonist delivery [91]. In a melanoma cell (B16)-injected C57BL/6 mouse model, a delivery of artificially combined siRNA targeting programmed cell death protein 1(PD-1-siRNA) and cytosine-phosphate-guanine oligodeoxynucleoties (CpG ODN) effectively reduced tumor size, and exhibited the longest survival, as compared with any single delivery of either siRNA or CpG ODN TLR agonist. In this group, the expression of cyclin D1, p-STAT3, and MMP2 decreased, while cleaved caspase 3 increased markedly. Furthermore, the recruitment of immune cells to tumor sites and the number of CD8+ T cells increased in mouse spleens, along with increased upregulation of TNF-α and IL-6.

Furthermore, phospholipids could also be incorporated in an amphiphilic conjugate for enhanced CpG-based TLR9 agonist delivery [92]. Self-assembly of pyridyldithiol-activated 1,2-dioleoyl-sn-glycero-3-phosphoethanolamine (DOPE) was conjugated with thiol-terminated hydrophilic CpG ODN, resulting in a self-adjuvating biomimetic anti-tumor nanovaccine. During the sustained and controlled long-term antigen supply, effective initial antigen stimulation and antigen cross-presentation via the MHC-1 pathway initiating CD8+T cell responses were observed. Consequently, in vivo examination of the therapeutic efficacy of the anti-tumor nanovaccine in EG7-OVA tumor models demonstrated the promotion of necrosis and apoptosis in cancer cells in tumor site, a significant tumor suppressive effect, and survival prolongation.

A conventional cytotoxic chemotherapeutic agent could also be used with the co-delivery with CpG TLR agonist for improved vaccination and tumor treatment [93]. To treat colon adenocarcinoma tumor, a dual delivery of docetaxel (DTX) and cholesterol-modified CpG 1826 using synthetic high-density lipoprotein nanoparticles (sHDL) has been suggested. Overexpressed SR-B1 in MC38 cells guided efficient in vitro cellular uptake of sHDL, and cytotoxic effect of DTX chemo-drug was maintained by particle formulation. *In vivo* outcome of sHDL using C56/BL6 mice also demonstrated that the combination of immunostimulatory agents effectively facilitated antitumor effects, and prolonged survival rate.

### 4.2. Synergistic Immunogenic Effects via Co-Delivery Using Diverse Co-Factors

Furthermore, the combinatorial exogenous delivery of triple immunostimulatory agents has recently also been investigated. For example, along with the dual TLR agonists of CpG and MPLA, tumor antigen B16 melanoma derived-TRP2 peptide could be incorporated in mesoporous silicon vector (MSV) microparticles [94]. Liposomal encapsulation of these stimulants was subsequently combined into MSV (MSV/TRP2-CM), and a sustainable release of cargos was obtained by silicon hydrolysis. The increment of antigen presentation efficiency by TLR agonists in DC cells has been detected, and immunization using MSV/TRP2-CM was found to be superior in T cell activity and IL-6 generation to any single component treatment.

On top of multiple activation by dual TLR agonists, synergistic immune stimulation of triple poly(I:C, pIC) (TLR 3 agonist for the reprograming of acute inflammatory state), R848 (TLR 7/8 agonist for inhibiting tumor-mediated T cell senescence), and macrophage inflammatory protein-3 alpha (MIP3α) (CCL20 for attracting cells expressing CCR6/CD196) in PLGA nanoparticles were also demonstrated [95]. In both models of TC-1 lung carcinoma and RMA T cell lymphoma, intra-tumoral administration of this triple nano-complex could indicate a functional therapeutic vaccination efficacy.

### 4.3. Clinical Relevance of TLR Agonist Delivery for Cancer Treatment

Even though previous studies for TLR agonist delivery using various biomaterial-mediated platforms have been intensively conducted, most recent clinical trials have focused on the antitumor efficacy itself of either a single TLR agonist or combinative administration with other co-factors. As of August 2021, on the basis of a literature search through www.clinicaltrials.gov (search keywords: ‘Cancer’ and ‘TLR agonist’), 48 studies were found (Appendix A). Among these trials, 16 studies had been completed, and 8 studies had been terminated. Among these 24 studies, a total of 6 of studies (4 in phase I and 2 in phase II) had been executed on the medication of a single TLR agonist. In addition, the other 18 studies (11 in phase I and 7 in phase II) were combinatory administration using TLR agonist along with other factors such as antibodies or chemo-drugs.

In particular, only 2 studies out of 48 were found that were related with the use of biomaterial-based delivery of TLR agonists (Table 5). Virus-like particles (VLPs) are molecules that resemble viruses but have no infections, due to the exclusion of the genetic material of virus. At the same time, VLPs display high-density repeat viral surface proteins that interact with innate immune system [96]. According to their immunological effectiveness, VLPs not only serve as the vaccine for the control of COVID-19, but also as therapeutic vaccines encapsulating CMP-001 [97,98].

## 5. Conclusions

According to the recently reported investigations, a series of biomaterial-based delivery platforms have shown facilitated activation of immune responses via exogenous TLR agonist delivery and associated synergistic therapeutic advantages of co-delivered immunomodulatory and biochemical stimulants. Various TLR agonists could be delivered via a series of biomaterial-mediated platforms such as liposomes, dendrimers, polyplex, stimuli-responsive polymeric particles, hydrogels, or lipoprotein-based scaffolds. Such delivery platforms could provide (1) the inhibition of rapid diffusional dissipation from the local injected site, (2) the protection of cargo TLR agonists from harsh surrounding environments, and (3) the effectiveness in simultaneous co-delivery of multiple factors. Either a single form of TLR agonist or a multiple combination of TLR agonists with the aid of supporting immunogenic co-factors demonstrated a successful activation of APCs, enhanced cytokine secretion, and up-regulated the interplays of immune cellular responses, T cell activation, and vaccination-mediated tumor suppression. Therefore, the optimization in the selection of delivery methods and the cargo combinations should be considered for engineering TLR agonist delivery. Even though there is a need to develop more clinically associated translational medicines based on TLR signaling cascades, it could be reasonably speculated that exogenous delivery of combinational TLR agonists is a promising immunotherapeutic strategy to treat cancers.

## Figures and Tables

**Figure 1 pharmaceutics-13-01374-f001:**
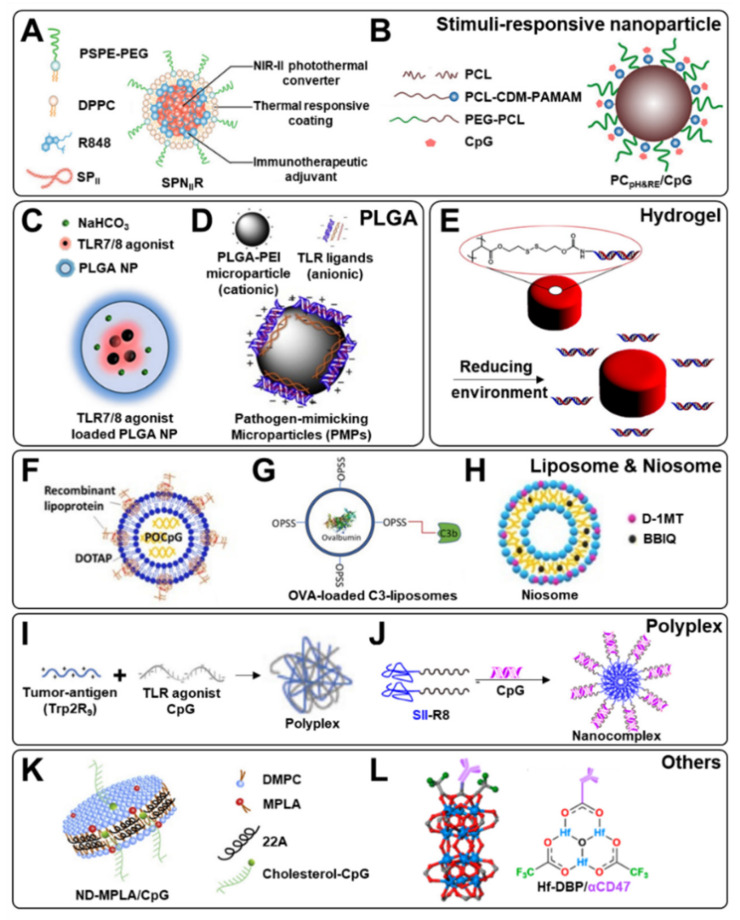
Delivery of TLR agonists using various platforms. (**A**,**B**) Exogenous TLR agonist delivery via stimuli-responsive vehicles/platforms could effectively release the cargos at the target tumor, and synergistically active release could be obtained with other treatments (e.g., PTT) ((**A**) reproduced with permission from ref. [34], 2021 Wiley Online Library; (**B**) reproduced with permission from ref. [35]. 2020 American Chemical Society), (**C**,**D**) PLGA nanoparticle-mediated delivery ((**C**) reproduced with permission from ref. [36]. 2020 American Chemical Society; (**D**) reproduced with permission from ref. [37]. 2014 Elsevier), (**E**) PRINT hydrogel for local tumor delivery of CpG (Reproduced with permission from ref. [38]. 2012 American Chemical Society, (**F**–**H**) Liposome/niosome for the facilitated encapsulation of TLR agonists and specific targeting efficacy ((**F**) reproduced with permission from ref. [39]. 2020 Multidisciplinary Digital Publishing Institute; (**G**) reproduced with permission from ref. [40]. 2019 Elsevier; (**H**) reproduced with permission from ref. [41]. 2020 Elsevier), (**I**,**J**) polyplex-type platform for RNA/DNA types of TLR agonists ((**I**) reproduced with permission from ref. [42], 2020 Frontiers; (**J**) reproduced with permission from ref. [43]. 2020 Elsevier), (**K**) Synthetic high-density lipoprotein nanodiscs or (**L**) metal–organic framework could be also applied ((**K**) reproduced with permission from ref. [44]. 2018 Elsevier; (**L**) reproduced with permission from ref. [45]. 2020 American Chemical Society).

**Figure 2 pharmaceutics-13-01374-f002:**
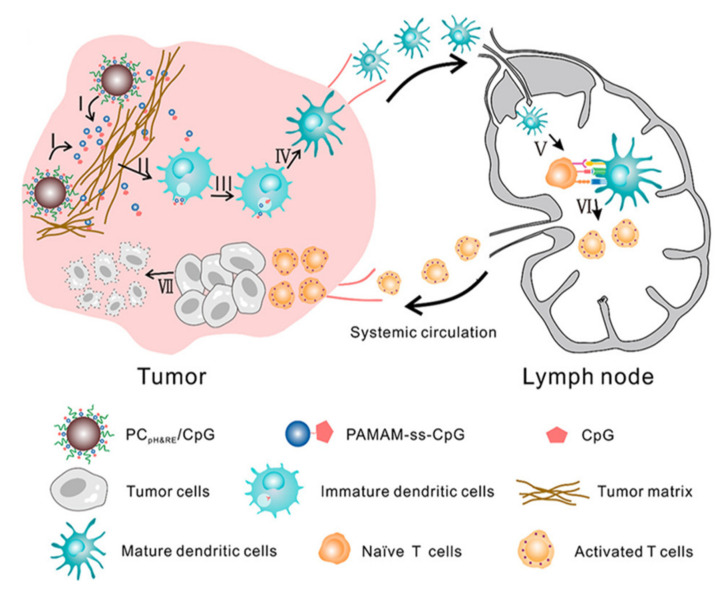
Schematic illustration of PAMAM nanocarrier to deliver TLR 9 agonist for cancer immunotherapy. I: PAMAM-ss-CpG release from nanocarrier in acidic tumor microenvironment. II: Uptake of PAMAM-ss-CpG by DCs. III: CpG release from PAMAM by reduction of the disulfide bond in intracellular environment. IV: Activation of DCs by CpG, V: Translocation of DCs toward lymph nodes. VI: Naïve T cell activation by matured DCs. VII: Facilitated infiltration into tumor tissues and anti-tumor effect of CD8^+^ T cell. Reproduced with permission from ref. [35]. 2020 American Chemical Society.

**Figure 3 pharmaceutics-13-01374-f003:**
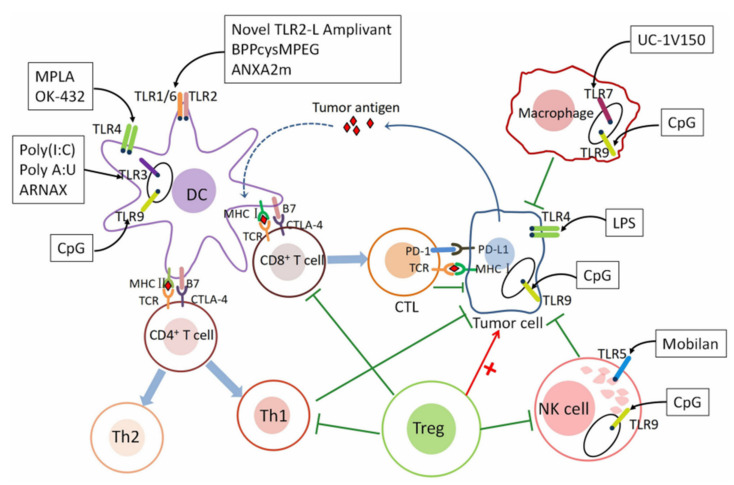
Scheme of TLR agonist-mediated signaling cascades in tumor immunity and immune check point pathways. Reproduced with permission from ref. [50]. 2020 Frontiers.

**Figure 4 pharmaceutics-13-01374-f004:**
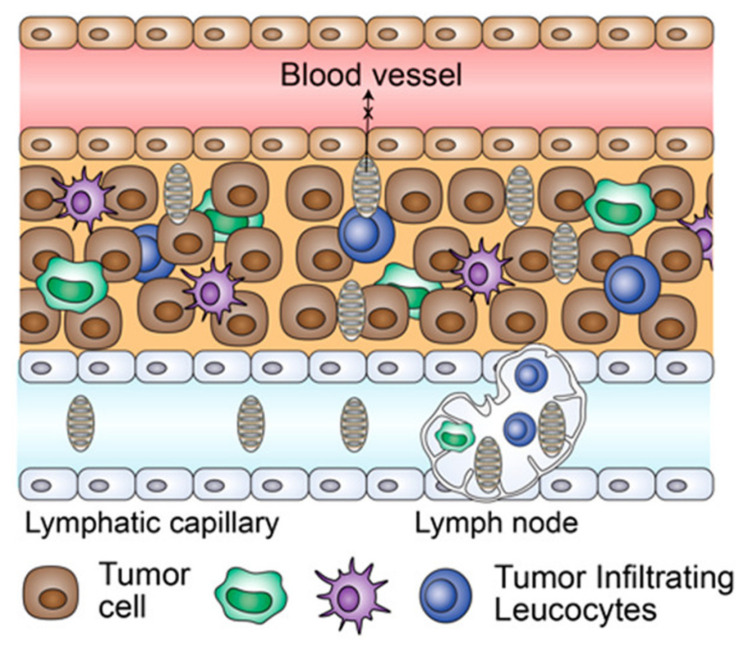
Multicomponent TLR agonist assembly (MTA) treatment could enhance the migration and mobility of tumor-infiltrating leukocytes (TIL) to tumor microenvironment. Reproduced with permission from ref. [82]. 2020 American Chemical Society.

**Figure 5 pharmaceutics-13-01374-f005:**
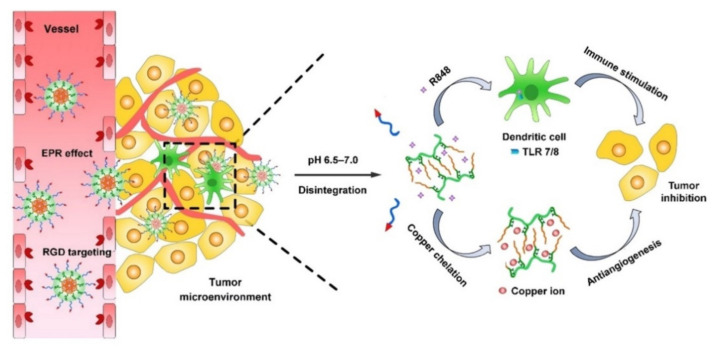
Combinative anti-tumor effect of R848 and TETA-DTC copper chelator. Reproduced with permission from ref. [90]. 2019 Elsevier.

**Figure 6 pharmaceutics-13-01374-f006:**
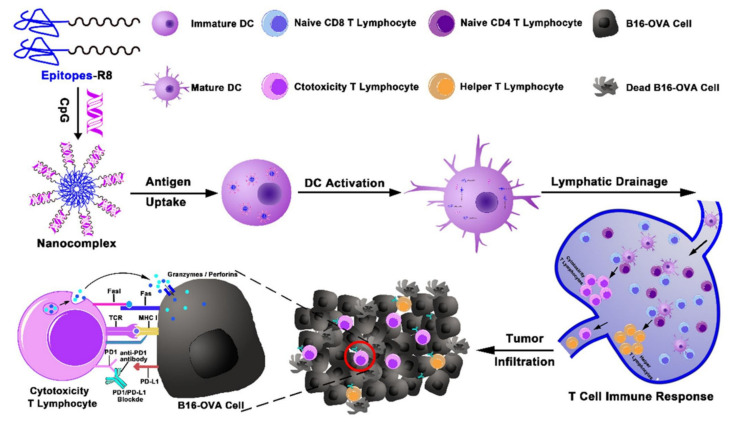
DC activation via nanocomplex consisted of epitopes-R8 and TLR agonist and subsequent immune response in T cells for augmented cancer immunotherapy. Reproduced with permission from ref. [43]. 2020 Elsevier.

**Table 1 pharmaceutics-13-01374-t001:** Biomaterial-mediated delivery platforms for exogenous single TLR agonist delivery.

TLR	TLR Agonist	Delivery Platform	Target Cancer	Ref
Mechanism		Improvements		
7/8	GU-rich ORN	RNA/DNA hydrogel	Colorectal cancer	[31]
Slow cargo release from a hydrogel platfrom DC uptake of released RNA cargo Enhanced release of cytokines (TNF-α, IL-12) by DCs Activation of CTLs and induced cancer cell apoptosis by cytokines		Limited uptake of endocytic RNA by APCs Hydrolysis and clearance of cargo RNA by the RNase in the administration site Tetrapod RD3 improved biological stability and efficient uptake of cargos Facilitated immunostimulatory activity and antigen presentation		
7/8	R848	Hydrophobic PANI with hydrophilic GCS	Colorectal cancer	[32]
NP uptake by DCs and loaded R848 activates TLR7/8 Upregulation of co-stimulatory factors CD80 and CD86 Matured DCs secreted multiple pro-inflammatory cytokines (IL-6, TNF-α) Regulated anti-tumor immune responses Stimulated tumor-specific T cell response		Disruption of immune cells and subsequently weak immune responses by local high temperatures during photothermal processes PANI-mediated hyperthermia induced by relatively low temperature Upregulation of DAMP (HSP70) expression improve anti-tumor immunity Combination with R848 TLR agonist induced stronger anti-tumor immunity		
7/8	R848	SPN_II_R	Breast cancer	[34]
SPN_II_R induced local temperature increases by photoirradiation Deformation of lipid shell of SPN_II_R by hear generation and release of cargo R848 in tumor sites Promote DC maturation and expansion of antigen-specific CTLs		Possibility of severe irAEs by systemic distribution of R848 agonists Thermally responsive lipid shell for on-demand release of R848 in tumors		
7/8	Imidazoquinoline-based agonist 522	PLGA	Melanoma Bladder cancer	[46]
Efficient internaliztion into APCs Enhanced CD86 expression in BMDCs Initiation of more potent T cell responses Increased antigen (OVA)-presentation in BMDCs		Rapid clearance of soluble drug from injection site Encapsulation of TLR agonists in NPs improved a delivery of agonist payloads to DCs		
7/8	Imidazoquinoline-based agonist 522	Acidic pH-responsive PLGA	Melanoma	[47]
Acidic pH-responsive generation of CO_2_ gas in PLGA NPs· Rapid release of 522 by mechanical discruption of a polymeric shell Released 522 could induce DC activation and proliferation of IFN-γ+CD4+ T cells Subsequent activation of CD8+ T cells and NK cells		Rapid clearance of small cargo molecules from dermal layers. Improved cargo agonist encapsulation and endo-lysosome specific agonist release by using Acidic pH responsive PLGA NPs		
7/8	Imidazoquinoline-based agonist 522	Acidic pH-responsive PLGA	Lung cancer, Mammary carcinoma, Leukemia	[36]
NPs release TLR7/8 agonists at endo–lysosomes Increase CD70, CD80/86 and IL-12, IL-18 on BMDCs Induced IFN-γ and TNF-α Promote NK cells and Th1 immunity		Reduced effectiveness of TLR agonists through SC and IM delivery Acidic pH-responsive PLGA NPs achieve endo/lysosomal specific release of TLR7/8 agonists.		
9	CpG ODN	PAMAM dendrimers	Pancreatic cancer	[35]
Robust DC activation & increased CTL infiltration into tumors and TDLNs Increased the expression of CD80 and CD40 on BMDCs Increased the secretion of IL-6, IL-12p70, TNF-α on BMDCs Activation of CD8+ T cells		Unfavorable pharmacokinetic/biodistribution profiles of TLR agonist Poor intracellular uptake of TLR agonist Protection of cargo CpG ODN from nuclease degradation by nanocarriers Improved delivery of CpG ODNs into TIDCs using PAMAM dendrimers		
9	CpG ODN	PRINT nanoparticles	Lung cancer	[48]
PRINT-CpG increased the levels of TNF-α, IFN-γ, Fpr2, IL-12p40, IL-6, and CXCL10 in the treated lung Activated APCs triggered the activation of NK and T cells		Subcutaneous delivery of TLR9 agonist has no significant improvements Lung delivery of soluble CpG cause lung injury and resulting in liver damage. Appropriate formulation of CpG ODN using PRINT particles may reduce its local and systemic toxicity without lung inflammation		

ORN, Oligoribonucleotide; DC, Dendritic cell; TNF-α, Tumor necrosis factor-α; IL, Interleukin; CTL, Cytotoxic T lymphocyte; R848, Resiquimod; PANI, Polyaniline; GCS, Glycol-chitosan; CD, Cluster of differentiation; DAMP, Damage-associated molecular pattern; HSP, Heat shock proteins; TAA, Tumor-associated antigens; APC, Antigen-presenting cell; SPN_II_R, Semiconducting polymer nanoadjuvant; irAE, Immune-related adverse event; PLGA, Poly(lactic-co-glycolic acid); BMDC, Bone marrow dendritic cell; OVA, ovalbumin; IFN-γ, Interferon-γ; SC, Subcutaneous; IM, Intramuscular; NP, Nanoparticle; NK cell, Natural killer cell; ODN, Oligodeoxyribonucleotide; PAMAM, Polyamidoamine; TDLN, Tumor-draining lymph node; PRINT, Particle replication in non-wetting template; Fpr2, Formyl-peptide receptor 2; CXCL10, Chemokine interferon-γ inducible protein 10 kDa.

**Table 2 pharmaceutics-13-01374-t002:** Representative FDA-approved TLR agonists for cancer therapy.

Agent/ Source	Target Agonist	Cancer Type	Induction of Immune Responses	Mediation Step for Antitumor Outcomes	Possible Drawback	Ref
BCG/ *Mycobacteri**-um bovis*	2/4	Bladder cancer	(1) Mediates the antigen processing functions of APCs (2) Increases the surface expression of MHC class I in urothelial tumor cells (3) (3) Secretion of cytokines such as IFN-γ and IL-2 and antigen-specific T cell immunity	(1) Th1 cell-mediated (or acquired) immunity via CD+ F24+ T-cells and CD8+ CTLs (2) Th2cell-mediated (innate) immunity via activation of NK cells	(1) Possible BCG-associated side effects (2) Insufficient efficacy except superficial transitional cell carcinoma of the bladder	[67,73,74,75]
MPLA/ *Salmonella minnesota*	2/4	Cervical cancer	(1) Induce IFN-γ and TNF-α	(1) Increase in total active T cells and a decrease in monocytes (2) Induced significantly higher frequencies of antigen-specific memory B-cells and T-cells	Fever, chills, and rigor, bronchospasm, hypotensive	[67,76]
Imiquimod/ Imidazoqui-noline	7	Breast cancer	(1) Increases immunostimulatory cytokines including IFN-α, TNF-α, IL-1β and IL-6 (2) Activates innate immunity associated with type I interferon production (3) Induces secretion of proinflammatory cytokines, predominantly IFN-α, IL-12, and TNF-α (4) (4) Enhances DC maturation and antigen presentation	(1) CCL2-dependent recruitment of plasmacytoid DCs into the tumor (2) Enhanced IFN-γ-producing CD4 cells (3) Enhanced functional antigen-specific CD8 responses (4) Activation of Th1 and Tc1 T-cell responses	(1) Side effects of fatigue, malaise, fever, headache, and lymphocytopenia (2) Cause flu-like symptoms (3) Depression on treatment	[59,69,70]

BCG, Bacillus Calmette-Guerin; APC, Antigen-presenting cell; CD, Cluster of differentiation; MHC, Major histocompatibility complex; CTL, Cytotoxic T lymphocyte; IFN, Interferon; TNF-α, Tumor necrosis factor-α; CCL2, Chemokine (C-C motif) ligand; IL, Interleukin.

**Table 3 pharmaceutics-13-01374-t003:** Strategies in multiple TLR agonist delivery.

TLR	TLR Agonist	Delivery Platform	Target Cancer	Ref
Mechanism		Improvements		
2, 9	rlipoE7m, CpG ODN	DOTAP liposome	Lung cancer	[39]
DOTAP is successful in targeting DCs and · Activate DCs via the ROS pathwayInduce a Th1-biased immune response by IL-12 production IL-1 production and IL-10 reduction Increase CTLs and decrease Tregs		DOTAP prolong the half-life of CpG DOTAP increase delivery efficiency of CpG		
2/6_7	Pam2CSK4C-PEG4-DBCO-2Bxy	Amphiphile OL-DSPOE micelles	Melanoma	[82]
MTA induced a higher secretion of IFN-β and IL-12 Promote CD8+ Tcell and NK cell mediated immunity Significantly enhanced TIL Increased infiltration of CD8+ CTLs and NK cells		Low efficacy and unacceptable levels of off-target toxicity by rapid systemic diffusion of cargo TLR agonists Micellular MTA formulation did not induce significant systemic cytokine secretion with a enhnaced localization		
1/2, 4, 7/8, 9	Pam3CSK4, MPLA. R837, CpG ODN	PLGA particles	-	[83]
Increased expression of IL12p70 in BMDCs· Increased population of CD40+/CD86+ BMDCs Optimal particle formulation facilitated DC maturation wirh enhanced antigen cross presentation.		Easy diffusion & dissipation of small molecule adjuvants Subsequent cytokine surge caused severe adverse immune-toxicity effects, limited effective dose, and prevention of optimal interaction Unique immune polarization could be obtained by controlled delivery of triple combination of TLR agonists using PLGA		
4, 7/8, 9	MPLA, R848, CpG 1826	C3 liposomes	Breast cancer	[84]
Liposomal delivery of TLR agonists increased expression of inflammatory cytokines and activation markers Significant increase in gene expression of pro-inflammatory factors Increased number of monocytes and macrophages		Treatment specific immunogenic tumor antigen has many variables depending on the patient. C3-liposomes to deliver TLR agonist compounds to activate APCs without specific immunogenic tumor antigen		
4, 9	MPLA, CpG ODN	sHDL	Melanoma, Lung cancer	[44]
Increased expression of CD80, CD86 and IL-12p70 in BMDCs Effective in vitro activation of DCs Robust antigen-specific CD8+ T cell responses and antigen-specific CTL responses Regression of established tumors		Practical need on the optimal methodology for potent immune activation with combinations of TLR agonists TLR agonist co-loaded efficiently into sHDL TLR agonist loaded sHDL readily combined with a variety of subunit antigens		

rlipoE7m, Lipidated human papillomavirus E7 inactive mutant; ODN, Oligodeoxynucleotide; DOTAP, 1,2-Dioleoyloxy-3-trimethylammonium propane; DC, Dendritic cell; ROS, Reactive oxygen species; IL, Interleukin; CTL, Cytotoxic T lymphocyte; OL-DSPOE, Olyel-deprotected sugar poly(orthoester); CD, Cluster of differentiation; NK cell, Natural killer cell; MTA, Multicomponent TLR Agonist Assembly; TIL, Tumor infiltrating leukocytes; MPLA, Monophosphoryl lipid A; R837, Imiquimod; PLGA, Poly(lactic-co-glyocolic acid); PLP, Pathogen-like particle; R848, Resiquimod; C3, Complement component 3; TNF-α, Tumor necrosis factor-α; IRF7, Interferon regulatory factor 7; IP-10, Interferon γ-induced protein 10 kDa; APC, Antigen-presenting cell; sHDL, Synthetic high-density lipoprotein nanodiscs.

**Table 4 pharmaceutics-13-01374-t004:** Combinative delivery of TLR agonists with various co-factors.

TLR	TLR Agonist	Co-Factor	Delivery Platform	Target Cancer	Ref
Mechanism			Improvements		
7	BBIQ	D-1MT	Niosome	-	[41]
Stimulation of both Th1 and Th2-type immune responses by BBIQ Superior antitumor activity & reversing suppression of T cells by D-1MT			Long shelf life & higher stability of cargos by using BBIQ/D-1MT loaded niosomes		
7	IMD	Anti-CD47	Hf-DBP nMOFs	Colorectal cancer	[45]
Sustained release of IMD TLR agonist in vivo Repolarization of M2 to M1 macrophages by IMD Improving phagocytosis by blocking “don’t-eat-me” signal on tumor cells by αCD47			Enabled loading of αCD47 by Hf-DBP Ideal maintenance of a high local IMD concentration by nMOFs Simultaneous photodynamic and photosensitizing functions by nMOFs Effective eradication of tumors by nMOF-mediated cargo formulation		
7/8	R848	RGD-PEG-PGA	RPTDH (Copper chelating polymer)	Breast cancer	[90]
Acidic pH-responsive cleavege in the backbone of RPTDH/R848 nanoparticles. Stimulation of antitumor immunity via triggering DC activation and enhanced CD3 & CD8 expressions on T cells by R848 TLR agonist delivery Copper chelation inducing reduction of IL-1α expressions & suppression of NF-κB nuclear translocation on HUVECs and MDA-MB-231 cells			pH-sensitive copper-chelating function of RPTDH incuced in vitro antiangiogenic activity and inhibition of breast cancer cell growth Enhanced targeting and suppresive capacity for breast tumor and lung metastases by RPTDH/R848 nanoparticles		
9	CpG ODN	SVYDFFVWL peptide antigen	Polyplex-like NPs	Melanoma	[42]
Internalization of polyplexes into DCs initiated immunostimulatory signaling cascades Modulation in proliferation & immune function of T cells by altering the antigen composition in polyplexes			Tunability of polyplexes by interchanging Trp2 peptides Polyplex-mediated protection of cargo CpG TLR agonist Synergistic enhancement in DC activation, cytokine secretion, and tumor xenograft animal survival by Trp2R_9_/CpG polyplexes than a signle use of peptide antigen		
9	CpG ODN	Cell-penetrating peptide (R8)-conjugated melanoma specific peptide epitope	Peptide/CpG nanocomplex	Melanoma	[43]
Endocytosis of SII-R8/CpG nanocomplex by DCs within 1 hr. DC activation through MyD88-dependent TLR signaling pathways, subsequently provoking CTL responses to kill tumor cells			Promoted expressions of DC surface markers by the conjugation of R8 Augmented antitumor T cell responses by nanocomplex-stimulated DCs Effective vacination & synergy with PD-1 blockade therapy		
9	CpG ODN	PD-1-siRNA	attenuated *Salmonella*	Melanoma	[91]
Stimulation of plasma-like DCs producing IFN-α by CpG ODN CpG ODN increased immune responses by converting “cold” tumors to “hot” PD-1-siRNA decrease immunosuppression by inhibiting PD-1 signaling pathway Attenuated *Salmonella* could colonize and kill tumors and stimulate the immune system			The highest tumor reduction by co-delivery with PD-1-siRNA + CpG ODN Enhnaced recruitment of immune cells into tumor sites Upregulation of TNF-α & IL-6 in PD-1-siRNA + CpG ODN group		
9	CpG ODN	OVA antigen	DOPE-based amphiphiles	Lymphoma	[92]
Conjugation of DOPE phospholipids & CpG ODN Promoted endocytosis via various receptor-mediated pathways into DCs Antigen cross-presentation by the MHC-I pathway initiating CD8+ T-cell responses			Construction of self-assembled nanovaccine using DOPE-S-S-CpG ODN and OVA antigen Effective initial antigen stimulation & sustained long-term antigen supply Significant tumor suppression and survival prolongation in vivo by the dual cargo delivery in nanovaccines		
9	CpG ODN	DTX chemo-drug	sHDL	Colorectal cancer	[93]
Recognition of sHDL nanoparticles by endogenous HDL receptor scavenger receptor B1overexpressed in most cancers Activation & maturation of DCs Differentiation of B cells by the reception of ODN on a plethora of immune cells Subsequent cross-presentation of tumor-specific antigens & secretion of anti-tumor antibodies			Effective dual delivery of hydrophobic drug and vaccine antigen Significantly reduced tumor growth in mice & improved animal survivial treated with DTX-sHDL/CpG than a single DTX treatment Synergistic inhibition of cancer cell proliferation by DTX		
4, 9	CpG ODN, MPLA	TRP2 peptide	MSV microparticles	Melanoma	[94]
Delivery of TRP2 peptide, CpG, and MPLA through DC phagocytosis T cell response by presenting peptide antigen through MHC complex Activation of NF-κB signaling to produce TNF-α & IL-6			Protection of cargo TRP2 peptide by MSV microparticle (96 hrs) Persistent T cell activity & sustained IL-6 production by MSV-mediated multiple component delivery· Increased survival of immunized mice		
3, 7/8	Poly(I:C), R848	MIP3α	PLGA NPs	Lung cancer, Rhabdomyosarcoma	[95]
Enhanced expressions of CD40, CD80 and CD86 on DCs by multi-cargo uptake Production of IL-12 by DCs through the activation of endosomal TLR3 and TLR7/8			Effective intratumoral multi-cargo delivery Attraction of tumor-promoting immune-suppressive cells to the tumor microenvironment by MIP3α Improved survival of vaccinated mice by the combination of pIC, R848 and MIP3α		

BBIQ, 1-benzyl-2-butyl-1H-imidazo [4,5-c]quinolin-4-amine; Niosome, Non-ionic surfactant vesicles; D-1MT(or Indoximod), 1-methyl-D-tryptophan; R848, Resiquimod; DTC-TETA, Sodion triethylenetetramine-bisdithiocarbamate; RPTDH, RGD-PEG-b-PGA-g-(TETA-DTC-Phis); ODN, Oligodeoxynucleotides; TRP, Tyrosine-related protein; MPLA, Monophosphoryl lipid A; sHDL, Synthetic high-density lipoprotein nanoparticles; SLA, Leishmania; PLGA, Poly(lactic-co-glyocolic acid); MSV, Mesoporous silicon vector; TRIF, TIR-domaincontaining adapter-inducing interferon-β; TNF, Tumor necrosis factor; MyD88, Myeloid differentiation primary response gene 88; IRF7, Interferon regulatory factor 7; DOPE, 1,2-dioleoyl-sn-glycero-3-phosphoethanolamine; IMD, Imiquimod; DBP, Duffy binding protein; nMOFs, Nanoscale metal-organic frameworks; NP, Nanoparticle.

**Table 5 pharmaceutics-13-01374-t005:** Current clinical trials of TLR agonists using VLP delivery platform.

TLR	TLR Agonist	Phase	Co-Factors	Conditions	Status	Ref.
9	VLP-encapsulated CMP-001 (TLR 9 agonist)	II	Pembrolizumab	Melanoma of Unkown Primary and 5 more	Recruiting	NCT 04708418
		I, II	INCAGN01949	Locally Advanced Malignant Solid Neoplasm and 3 more	Not yet recruiting	NCT 04387071

VLP, Virus-like particles; AJCC, American Joint Committee on Cancer.

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
