# Peer review of "Engineering Therapeutic Strategies in Cancer Immunotherapy via Exogenous Delivery of Toll-like Receptor Agonists"

_pharmaceutics, 2021, doi:10.3390/pharmaceutics13091374_

Round 1

Reviewer 1 Report

I am impressed by the review - it is of course worth to publish in Pharmaceutics.

I would suggest to augment the tables with column explaining the mechanism of action and/or the main "obstacle" which is overcome by delivery system.

The index of abbreviations is desired.

Author Response

Please find the attached  file for the rebuttal.

Reviewer 2 Report

In the present review, Jeong et al, focus on the clinical development of TLR agonists as cancer therapy. The authors discussed the comparative efficacy of TLR agonist as free and in combination with nanobiomaterial-mediated delivery systems. Here in, the authors describe the use of biomaterial-based delivery platform for effective TLR agonist transport and the associated cooperative biomedical advantages. This review has been well written and the recent literatures were cited appropriately. However, there are some sections that need to be discussed in more in depth.

Comments:

  1. Authors may discuss about the TLRs agonists and their roles in immune checkpoints pathways, and how nanobiomaterial-mediated delivery system can improve the immune checkpoints mediated signalling pathways to invade the tumor cells.
  2. Authors may cite some references based on clinical trials related to nanobiomaterial-mediated TLRs agonists delivery system in cancer therapy.
  3. Authors should consider to discuss the importance of TLRs agonists as adjuvants for cancer vaccines, e.g. they are potent DC activators.
  4. Figure legend 2: Numbering of figure B missing.

Author Response

(The authors gave the same response as above.)

Reviewer 3 Report

Re: review

“Engineering Therapeutic Strategies in Immuno-Cancer Therapy via Exogenous Delivery of Toll-like Receptor Agonists” - Pharmaceutics

This manuscript intends to update the literature related to the delivery platforms for toll-like receptors (TLRs) agonists to improve the efficacy of this class of immunotherapeutic drugs while reducing their side effects.  While this approach is reasonable and potentially beneficial for cancer therapy, this report suffers from a number of flaws, which along with an impaired English language, makes the manuscript difficult to read.

First of all, the manuscript looks as an immunology review, not related to bioengineering. I was not able to identify any work from the authors to attest their expertise in immunology. This is a very important issue since not-understanding well a phenomena and presenting it in a different way may lead to impaired basic information. For example, the authors state in the abstract “the maturation of these DCs affect T lymphocyte differentiation”. The correct way for a reader with basic immunology knowledge should be “the maturation of DCs stimulate naïve T cells to differentiate into functional cells”. Moreover, it is strongly recommended that the authors to cite themselves for the purpose of showing expertise in the fields. The authors have not only never published immunology-related work, but they also have no work related to bioengineering of TLR agonists.

The manuscript is enumerative missing to show the big picture; it looks like a list of studies without any logic integration between topics and chapters.

There are so many details taken from the cited original articles, which are completely irrelevant for the main purpose of the manuscript.

Authors introduce words which are not used in academic text (like in the title “immune-cancer therapy”).

I noticed another unusual thing for a review, the multitude of figures which were identically copied from other articles (despite having author’s approval). In general, these figures should be modified, adapted to the purpose of the review with the author’s agreement.

The authors must discuss in detail the challenges, pitfalls and alternatives of these delivery platforms. This should probably be the most important section of this work.

Author Response

(The authors gave the same response as above.)

Round 2

Reviewer 3 Report

I would like to thank the authors for their outstanding work. The revised manuscript is responsive to my previous review comments including adding new information and comprehensive integration of these new information. I believe that this updated manuscript is suitable for publication now.